# Protein Succinylation and Malonylation as Potential Biomarkers in Schizophrenia

**DOI:** 10.3390/jpm12091408

**Published:** 2022-08-30

**Authors:** Bradley Joseph Smith, Caroline Brandão-Teles, Giuliana S. Zuccoli, Guilherme Reis-de-Oliveira, Mariana Fioramonte, Verônica M. Saia-Cereda, Daniel Martins-de-Souza

**Affiliations:** 1Laboratory of Neuroproteomics, Institute of Biology, Department of Biochemistry and Tissue Biology, University of Campinas, Campinas 13083-862, Brazil; 2Instituto Nacional de Biomarcadores em Neuropsiquiatria (INBION), Conselho Nacional de Desenvolvimento Científico e Tecnológico, São Paulo 05403-000, Brazil; 3Experimental Medicine Research Cluster (EMRC), University of Campinas, Campinas 13083-862, Brazil; 4D’Or Institute for Research and Education (IDOR), São Paulo 04501-000, Brazil

**Keywords:** succinylation, malonylation, schizophrenia, antipsychotics, biomarkers, post-translational modifications, shotgun proteomics

## Abstract

Two protein post-translational modifications, lysine succinylation and malonylation, are implicated in protein regulation, glycolysis, and energy metabolism. The precursors of these modifications, succinyl-CoA and malonyl-CoA, are key players in central metabolic processes. Both modification profiles have been proven to be responsive to metabolic stimuli, such as hypoxia. As mitochondrial dysfunction and metabolic dysregulation are implicated in schizophrenia and other psychiatric illnesses, these modification profiles have the potential to reveal yet another layer of protein regulation and can furthermore represent targets for biomarkers that are indicative of disease as well as its progression and treatment. In this work, data from shotgun mass spectrometry-based quantitative proteomics were compiled and analyzed to probe the succinylome and malonylome of postmortem brain tissue from patients with schizophrenia against controls and the human oligodendrocyte precursor cell line MO3.13 with the dizocilpine chemical model for schizophrenia, three antipsychotics, and co-treatments. Several changes in the succinylome and malonylome were seen in these comparisons, revealing these modifications to be a largely under-studied yet important form of protein regulation with broad potential applications.

## 1. Introduction

### 1.1. Schizophrenia and Antipsychotics

Schizophrenia (SCZ) is a complex and multifactorial illness that is estimated to affect between 0.3 and 0.7% of the population worldwide [1]. It is characterized by changes in an individual’s ability to perceive and react to the world around them and is brought on by a culmination of biological and environmental factors [2] that lead to the individual experiencing deficits in social appropriateness and hallucinations or psychosis [3], among other symptoms. Current treatment options for SCZ are based entirely on symptom management through antipsychotics (typical and atypical) and psychosocial therapy. Antipsychotic medications can be grouped into two overarching categories, named by their order of discovery: typical and atypical antipsychotics.

Typical antipsychotics (TAPs, also called first-generation antipsychotics) include chlorpromazine and haloperidol. TAPs bring about their therapeutic effects mainly by acting as antagonists of the dopamine D_2_ receptor (D_2_R) [4], and this reduction in D_2_R activity reduces positive symptoms in patients with SCZ. However, due to their binding kinetic profile, TAPs can cause extrapyramidal symptoms including tardive dyskinesia, a condition that affects a patient’s motor abilities [5]. As an alternative, atypical antipsychotics (AAPs), which include clozapine and risperidone, possess different binding kinetics, reducing the chances of developing extrapyramidal symptoms [6,7]. A more rapidly associating and dissociating binding profile is understood to be behind this improvement [8,9]. AAPs also bind to the serotonin 5-HT_2A_ receptor [10], reducing hyperprolactinemia [11] and improving cognition [12]. In contrast, AAPs can induce agranulocytosis [13], a serious, potentially lethal blood disorder if unmanaged [14]. Exactly how different binding profiles lead to changes in side effects and therapeutic ability in a given patient is still poorly understood [15], despite similar overall drug effectiveness [16], and is compounded by the complex and widely varying profiles of each medication.

### 1.2. Succinylation and Malonylation of Lysine

At the base of this study are two post-translational modifications (PTMs), both of which are covalent additions of an acyl group to a lysine residue. The addition of succinate, called succinylation, has been found across many domains of life, having been confirmed in *E. coli*, *S. cerevisiae*, HeLa, and mouse liver cells [17,18]. The proposed source of succinate for this addition is succinyl-CoA [17], a key intermediate in the TCA cycle, originally suggesting a link with metabolism and energy production. This link has since been confirmed in studies of site locations, showing that many mitochondrial and metabolic proteins are succinylated (2572 sites on 990 proteins) [17]. Further supporting this association with metabolism, the succinylome rapidly responds to hypoxia, changing in as little as 20 min [19]. This PTM occurs most often as a spontaneous modification; however, some lysine succinyltransferase activity has been observed by carnitine palmitoyltransferase 1A (CPT1A) [20] and lysine acetyltransferase 2A (KAT2A) [21]. Desuccinylation occurs primarily through the sirtuin family protein SIRT5 [22,23,24]. Deregulations in SIRT5 activity can modulate tumor growth and tumorigenesis and reduce fatty acid oxidation and ATP production [25].

The other PTM that is focused on in this study is lysine malonylation, occurring similarly to succinylation via the spontaneous transfer of malonate from malonyl-CoA [26]. Malonyl-CoA itself is an intermediate in fatty acid synthesis, and malonylation was therefore hypothesized to have strong ties with metabolism and energy regulation, a hypothesis that was supported by analytical and knockout studies [24,26]. Demalonylation occurs by the same sirtuin member SIRT5 as succinylation [22,23,24], suggesting an additional relationship between the two metabolic PTMs.

Both of these modifications, despite their broadly spontaneous nature, are not thought to occur without specificity nor unintentionally due to the ability to predict modification sites for both succinylation [27,28] and malonylation [29]. Linking antipsychotics with these modifications, weight gain from antipsychotics has been linked with perturbations in fatty acid oxidation, specifically malonyl-CoA [30]. Moreover, SCZ is a disorder that consistently presents metabolic and mitochondrial dysregulations [31,32,33,34,35,36], making the links between succinylation and malonylation with metabolism [19,23,26,37,38] especially relevant.

### 1.3. Models for Studying Schizophrenia and Antipsychotics

Postmortem brain tissue from patients with schizophrenia is a classic and frequently used source of omic data; however, it cannot undergo any treatments, changes, or modifications. This makes cell cultures a vital part of researching SCZ. Oligodendrocytes (OLDs) are one such model, since OLDs are cells that provide neurons with structure, nutrients, and protection via myelination. OLDs have been implicated in some of the dysregulations in and symptoms of SCZ [39,40,41,42,43], thereby reinforcing the link with myelin-associated protein dysfunctions observed in SCZ [44]. One human OLD precursor cell line, MO3.13, has been shown to stay in a maturity state that neither progresses nor differentiates [45], making it easier to study the effects of different insults and treatments on these cells.

Glutamatergic dysfunction has been seen in patients with SCZ, and dizocilpine (MK-801), an NMDA receptor antagonist, is a documented model for SCZ, simulating this dysfunction. This model is supported by behavioral studies in rats, showing the induction of positive [46], negative [46], and cognitive [47] symptoms, additionally causing neurochemical and neuropathological changes similar to those seen in SCZ patients [48,49]. Although MK-801’s induction of positive symptoms is thought to differ from the PCP model [46,50], it is still a documented pharmacological model for psychosis [51]. Exposing a cell culture to MK-801 as a model for SCZ has already been standardized and published [52].

In this study, RAW data files from several shotgun mass spectrometry studies were reanalyzed to identify and compare succinylation and malonylation sites. The postmortem brain tissue of patients with schizophrenia was analyzed against mentally healthy controls, consisting of the total cell lysates of the corpus callosum and cerebellum and mitochondrial enrichments of the posterior cingulate cortex, caudate nucleus, and cerebellum. The roles of the corpus callosum [53,54,55,56,57,58,59,60], cerebellum [61,62,63,64,65,66,67], posterior cingulate cortex [68,69,70,71,72,73], and caudate nucleus [74,75,76,77,78,79,80] have all been studied in regard to SCZ. MO3.13 cells were also studied, having been exposed to MK-801, an antipsychotic (haloperidol, chlorpromazine, and quetiapine), or a combination of the two.

This study aims to reveal dysregulations in proteins and pathways that are not otherwise visible through genomics, transcriptomics, or standard proteomic workflows. Highlighting these otherwise overlooked pathways could identify foci for further studies related to the development, progression, identification, or treatment of SCZ. In addition, PTM profiles can increase the knowledge of the therapeutic and side effects of antipsychotics.

## 2. Materials and Methods

### 2.1. Postmortem Tissue Preparation

The cerebellum whole lysate and the mitochondrial enrichment of the cerebellum, posterior cingulate cortex, the caudate nucleus [81], and the corpus callosum whole lysate [57] were extracted, prepared, and analyzed as previously described. All brain tissue samples were provided by the BrainNet Europe consortium and were previously collected following German Tissue Law. Tissue samples from patients with SCZ were collected at the State Mental Hospital, Wiesloch, Germany; control samples were collected at the Institute of Neuropathology, Heidelberg University, Heidelberg, Germany.

All patient assessments, postmortem evaluations, and collection procedures were approved by the Ethics Committee at the Faculty of Medicine, Heidelberg University, Germany (application received on 9 November 1998 and approved 1 December 1998) and the Ethics Committee of the Faculty of Medicine at the University Medical Center Göttingen, Germany (application number 19/4/11, approved on 21 June 2012), and carried out in accordance with the Declaration of Helsinki. More information about patient classification, the calculation of chlorpromazine equivalents, and patient diagnosis can be found as published by Saia-Cereda et al. [56]; donor information involving the samples analyzed in this study has been provided (Appendix A).

### 2.2. Cell Line Growth and Preparation

MO3.13 oligodendrocyte precursor cells were grown, treated, and lysed according to the protocol published by Brandão-Teles et al. [52]. In joint treatments, cells were incubated with MK-801 (final concentration 50 µM) for 4 h before the antipsychotic was added to the same medium (to a concentration of 50 µM) for an additional 4 h. MO3.13 cells were sourced from Cedarlane Cellutions Biosystems, Inc. (product code CLU301).

### 2.3. Liquid Chromatography-Mass Spectrometry

The methodology for sample injection and acquisition was published for the corpus callosum total lysate [57], cerebellum total lysate, and mitochondrial fractions of the posterior cingulate cortex, caudate nucleus, and cerebellum [81], all of which relied on 2D LC-HDMS^E^. MO3.13 cell lysates were prepared and injected according to a previously published protocol [52], utilizing 2D LC-UDMS^E^, ramping the collision energy from 25 to 55 eV. In all experiments, samples were digested with trypsin and, for each sample, 1 µg of peptides was injected into a nanoAcquity M-Class ultra-performance liquid chromatography (UPLC) system (Waters Corporation, Milford, MA, USA) coupled online to a Synapt G2-Si Q-ToF mass spectrometer (Waters Corporation, Milford, MA, USA) with ion mobility separation (IMS), operated in resolution mode, and ionized using electrospray ionization in positive mode (ESI+). Human Glu1-fibrinipeptide was used as the lock mass, sampled every 30 s. Other unmentioned instrument conditions may have varied between experiments; more details can be found in the methods section of the above-cited articles for each sample type. The mass spectrometry proteomics data have been compiled and deposited to the ProteomeXchange Consortium via the PRIDE [82] partner repository with the dataset identifier PXD014188.

### 2.4. Protein Identification

Progenesis QI for Proteomics (version 3.0.6039) aligned the extracted ion chromatograms (XICs) for each sample source before identifying the peptides and proteins in each sample and performing relative quantitation using the top 3 Hi-N method. Protein identification was performed using the following parameters: maximum ion charge of +8, trypsin cleaving sites (up to one missed cleavage, not counting modified lysine residues), a maximum protein mass of 600 kDa, fixed carbamidomethyl C, variable oxidation M, variable acylation (either succinylation or malonylation) K, a minimum of 2 fragments per peptide, 5 fragments per protein, and 1 unique peptide per protein, using the Uniprot revised *Homo sapiens* database (February 2018 for postmortem studies and October 2018 for MO3.13 studies). An FDR limit of 4% was used with an on-the-fly reversed list generated by Progenesis. Raw data from Progenesis were filtered for a peptide mass error < 20 ppm. A peptide containing the modification of interest with ANOVA ≤ 5% between groups for quantitation was considered differentially expressed and its constituent protein was identified by its Uniprot accession number.

### 2.5. Data Analysis

To identify the pathways enriched by the proteins to which these dysregulated sites of modification belonged, the Uniprot accession numbers were loaded into the Reactome Pathway Database [83]. Pathways with an FDR ≤ 1% and *p*-Value ≤ 0.01 (Student’s *t*-test) were considered enriched. Data visualization was performed with the ggplot2 package [84] in an R environment (v4.1.2). To categorize attenuation for antipsychotics, first, comparisons were made with the two vehicles (DMSO or HCl in Milli-Q water) to remove any differences caused by the vehicles and other variable proteins, whereupon only averaged quantitation scores between samples within 10% of each other were selected. Next, an average of the two controls was used as a baseline value to categorize all of the remaining conditions. Modifications that were dysregulated by 10, 25, or 50% in either direction before returning over that threshold towards control levels indicated mild, moderate, and severe dysregulation. A gene was then marked with an asterisk (*) for each additional threshold passed, with the proteins marked with (**) being those that were disturbed by over ±50% by MK-801 and subsequently returned to within 10% of their control levels upon treatment with the antipsychotic.

## 3. Results

### 3.1. Postmortem Samples

#### 3.1.1. Mitochondria-Enriched Protein Lysates

In mitochondrially enriched tissue samples from three brain regions, multiple proteins were found to have changes in succinylation or malonylation (ANOVA ≤ 0.05; Appendix A). No proteins with dysregulated sites were found in more than one region. TUFM (mitochondrial tu translation elongation factor), which plays a role in mitochondrial protein translation, was found to have both up- and downregulated sites of malonylation in the posterior cingulate cortex (PCC). The caudate nucleus (CN) had fewer changes than the other two samples despite having a similar number of identified modified peptides (Figure 1a).

All proteins, as well as upregulation and downregulation subgroups, for each PTM, were pooled from brain regions. This revealed some additional pathways (FDR ≤ 0.01; *p*-Value ≤ 0.05); however, still no pathways were identified in malonylation pools. Downregulated succinylation sites and the succinylation pool both identified “Detoxification of Reactive Oxygen Species” entities, and in the overall pool, “Cellular responses to stress” (Appendix A).

#### 3.1.2. Total Cell Lysates

The two brain regions that were studied presented opposing profiles in regard to the number of sites. The corpus callosum (CC) showed a decrease in both modifications, while the cerebellum (CER) exhibited an increase in these modifications (Figure 1b). Additionally, the CER had about half the number of differentially modified peptides as the CC, despite similar peptide identification numbers (Figure 1c).

Proteins related to heat shock response/heat stress response (two separate pathways but with the same constituent entities) and the excretion of clathrin-coated vesicles from Golgi bodies had downregulated sites of succinylation. Proteins with upregulated succinylation sites did not enrich for any pathways (FDR ≤ 4%). When up- and downregulated modification sites were combined, mitochondrial growth and development were highlighted, including cristae formation and mitochondrial biogenesis.

Several pathways were found to have lower levels of malonylated proteins in the total cell lysates, specifically membrane and cytoskeletal trafficking pathways as well as organelle biogenesis and protein glycosylation. Proteins with upregulated malonylation sites did not enrich for any pathways (FDR ≤ 4%). Combining up- and downregulation modification sites, only a few overall differences were observed: cell cycle, regulation apoptosis, and signaling pathways (Appendix A).

### 3.2. Expression Profiles in MO3.13

#### 3.2.1. MK-801

When MO3.13 human oligodendrocyte precursor cells were exposed to the NMDA receptor antagonist MK-801, several PTM sites exhibited changes (ANOVA ≤ 0.05; Appendix A) in both directions. When data were divided into up- or downregulated sites and analyzed with the Reactome Pathway Database, constituent pathways included RNA metabolism, splicing, translation, and ribosomal function, and the CCT/TriC pathway, a category of chaperonins responsible for cytoskeleton proteins [85].

#### 3.2.2. Antipsychotics

Each of the three antipsychotics affected the modification of proteins that subsequently enriched for biological pathways using the Reactome Pathway Database (Appendix A). Chlorpromazine and haloperidol induced marked changes in the succinylome and malonylome of MO3.13 cells with few changes occurring in response to quetiapine, for which a much smaller fraction of acylated peptides were found to be affected (Appendix A). Overall, haloperidol affected more biological pathways than chlorpromazine, increasing both succinylation and malonylation on proteins with strong enrichment values. Many pathways were shared between these two modifications, more specifically including many RNA metabolism, protein translation, axon guidance, stress response, and nonsense-mediated decay pathways.

#### 3.2.3. Antipsychotics and Succinylation

In the MO3.13 cells incubated with antipsychotics, several succinylation sites were affected (ANOVA ≤ 0.05; see Appendix A), and the modified proteins enriched for biological pathways with the Reactome Pathway Database (Appendix A). The overall number of sites affected by each antipsychotic varied greatly between antipsychotics: (up-/downregulated) haloperidol 230/169, chlorpromazine 95/70, and quetiapine 57/4. The pathway enrichment analysis did not return any enriched pathways for this quetiapine. The associated pathways identified by the Reactome Pathway Database were compiled for visualization (Figure 2).

#### 3.2.4. Antipsychotics and Malonylation

The MO3.13 cells incubated with antipsychotics also exhibited site-specific changes in malonylation (ANOVA ≤ 0.05; see Appendix A). As with succinylation, the number of dysregulated sites varied distinctly between antipsychotics, exhibiting a similar ratio of deregulated sites. The number of sites by haloperidol, chlorpromazine, and quetiapine (up/downregulated) were 280/156, 114/72, and 50/5, respectively. Due to the low number of sites affected by quetiapine, the pathway enrichment analysis returned very few pathways for this antipsychotic. Despite a lack of overlap of individual proteins between conditions, some biological pathways were affected by all three antipsychotics, specifically involving RNA metabolism and translation. The pathways associated with these changes were identified with the Reactome Pathway Database (Appendix A). The pathways were compiled for visualization (Figure 2).

#### 3.2.5. Attenuation Profiles of Succinylation and Malonylation

MO3.13 cells were incubated with MK-801 and then treated with one of three antipsychotics to highlight proteins and pathways that may be associated with the therapeutic effects of antipsychotics in unhealthy cells. The sites disturbed by MK-801 were classified into three groups based on their change in expression (Mild ±10%, Moderate ±25%, and Severe ±50%), and those that returned to control or near-control levels were considered attenuated. Sites that returned closer to control levels after a more severe dysregulation are marked with asterisks (Table 1).

Succinylation sites that were considered to exhibit partial or full attenuation by antipsychotics treatment were seen on proteins associated with RNA and translational control, the cytoskeleton, and metabolism. There was no overlap of individual proteins between the different antipsychotics tested. Attenuated malonylation sites were on proteins involved with metabolism, protein folding, secretory pathways, and the cytoskeleton, among other pathways.

The proteins that were most severely affected (>50% change from control levels) by MK-801 and returned to within 10% of their baseline (Table 1) were, for succinylation, SNRNP70 (small nuclear ribonucleoprotein U1 subunit 70; haloperidol), MATR3 (matrin-3; chlorpromazine), and RPS15 and CNP (40S ribosomal protein S15, 2′,3′-cyclic-nucleotide 3′-phosphodiesterase; quetiapine). For malonylation, these were ACLY, HIST1H1E, and ESD (ATP-citrate synthase, histone H1.4, S-formylglutathione hydrolase; chlorpromazine) and CLTCL1 and GNL3 (clathrin heavy chain 2, guanine nucleotide-binding protein-like 3; quetiapine).

## 4. Discussion

PTM levels were found dysregulated not only in postmortem tissue from patients with schizophrenia but also in MO3.13 cells exposed to the MK-801 model for schizophrenia and/or antipsychotics. The existence of changes in every condition reinforces the idea that both succinylation and malonylation can play important and distinct roles in protein post-translational regulation. Such considerable differences in the succinylome and malonylome under changing environmental conditions suggest that these PTMs require attention, not just the more widely studied modifications such as phosphorylation and acetylation. Even more importantly, the possible regulatory effects resulting from these PTMs raise the hypothesis that they may be valid targets for biomarker investigations for metabolic and neurological disorders such as schizophrenia.

Various proteins were found to be differentially modified in postmortem studies in more than one sample, and pathways related to metabolic and cytoskeletal proteins were recurrent. Both of these pathways are widely documented dysregulations in SCZ, psychosis, and antipsychotic treatment response at both cellular and systemic levels [31,34,86,87,88,89,90,91,92,93], manifesting primarily as dysregulations in mitochondrial function, glucose metabolism, the cytoskeleton, and myelination. Using these data as a point of reference and as proof of concept, the succinylome and malonylome of peripheral biofluids of patients with schizophrenia need to be included in future studies to determine their capacity to be diagnostic, prognostic, or treatment efficacy biomarkers.

Moreover, more research is needed to determine if these dysregulations could also be therapeutic targets. For example, actin polymerization is altered in anterior cingulate cortex samples from elderly patients with SCZ, despite normal protein and transcription levels [94]. Such dysregulation is suggested to drive certain side effects of various mental disorders, including schizophrenia [95]. This is further reinforced by relations between the actin cytoskeleton and NMDAR function [96,97,98], a receptor in one of the principal dysregulated neurotransmitter pathways studied in SCZ [99,100]. Since PTMs can regulate protein stability and protein–protein interactions, the dysregulations of the succinylome, involving metabolic and mitochondrial proteins, and malonylome, involving the cytoskeleton, principally cytoskeletal transport pathways (Appendix A), suggest a link with SCZ worth investigating more deeply.

When MO3.13 cells were exposed to MK-801, more differences (ANOVA ≤ 0.05) were seen than in the brain tissue, likely due to cell homogeneity, a reduction in exogenous factors, and potentially also the form of collection compared to postmortem samples, as hypoxia has been found to affect the succinylome in as little as 20 min [19]. In these cells, MK-801 induced changes in pathways involved with RNA processing and metabolism, translation, and protein trafficking and folding. While the dysregulations at the pathway level were similar in both MO3.13 cells and postmortem tissue, no individual proteins with dysregulated sites were seen. This could be due to confounding factors originating from patient backgrounds or the type of analysis since a single cell type was focused on rather than overall heterogeneous tissue expression. Yet another possibility is that MK-801 does not mimic the PTM changes seen in SCZ. More studies with cell cultures, co-cultures, cerebral organoids, and patient-derived tissue can help to confirm or refute these possibilities.

When the MO3.13 cells were exposed to antipsychotics alone, multiple changes in the succinylome and malonylome were also seen. Quetiapine did not affect the succinylation of sufficiently many proteins to enrich for biological pathways; however, proteins involved in RNA and protein metabolism, stress response, and axon guidance were affected by chlorpromazine and haloperidol. Malonylation was modulated on proteins involved in many of the same overall pathway groups as succinylation. Protein trafficking, localization, and folding were also affected in several conditions. Despite some overlaps in pathways, the succinylome and malonylome differed between antipsychotics (Figure 2), likely due to the different receptor-binding profiles of each antipsychotic. The investigation of more antipsychotics of different chemical classes could reveal the effects of each neurotransmitter receptor on these PTMs, and determining which PTMs are modified in response to each antipsychotic could assist in developing biomarkers for accompanying treatment adhesion and efficacy.

Quetiapine, haloperidol, and chlorpromazine all expressed varying degrees of PTM dysregulation (Appendix A), despite little change to overall protein expression. Between 5 and 8% of all quantified peptides were modified, which does not explain the 10-fold difference between the percentage of dysregulated peptides being modified in haloperidol vs. quetiapine (Appendix A). One possible cause is that quetiapine is the only second-generation (atypical) antipsychotic. Alternatively, different side effects resulting from varying receptor-binding profiles, such as extrapyramidal effects and metabolic syndrome, could be causing different downstream cellular dysregulations. This difference should be further investigated with other antipsychotics to investigate how to manage or reduce the undesirable side effects caused by each antipsychotic class.

When cells were incubated with MK-801 and subsequently treated with one of three antipsychotics, some PTM sites were seen to return to more control-like levels. These attenuated sites were on proteins largely associated with RNA- and DNA-binding, transcription and translation regulation, the cytoskeleton, metabolism, and protein transport. Studying these proteins in more depth could provide a great deal of insight into the mechanisms of action of antipsychotics, how symptoms manifest themselves, and how they are attenuated. Moreover, such PTMs would offer possibilities to empirically measure treatment efficacy or potentially predict which antipsychotic from a list would best attenuate the dysregulations stemming from SCZ in a given patient.

The changes observed in this study suggest an important cellular response to many stimuli with the capacity to affect metabolic pathways, transcription, protein metabolism, and the cytoskeleton. This calls attention to lysine succinylation and malonylation as important targets for proteomic studies. Further investigations can improve upon this preliminary work by analyzing and comparing not only overall expression but also the percent modification of individual PTM sites compared to the unmodified version. Studying the effects these PTMs have on proteins could help determine if the observed changes are behind the symptoms of SCZ, or rather if this is only a response to an upstream dysregulation. Additional studies can confirm the cause of the differences seen between antipsychotics and if these differences can be targeted to develop better medication, improve current options by reducing side effects, or generate biomarkers for disease diagnosis, prognosis, and treatment.

## 5. Conclusions

Post-translational modifications are a growing field of research within proteomics, and the vast number of known modifications provides a broad field of study. Due to the role of PTMs in many facets of protein regulation, especially protein and complex stability, enzymatic activity, and protein–protein interactions, they hold great potential for human health and disease. In this work, two post-translational modifications with inherent ties to metabolism were probed in the context of schizophrenia and its treatment to investigate their possible use as biomarkers, a relevant target due to the metabolic dysregulations seen in both the mental disorder and in the side effects of antipsychotic treatment. Several dysregulations were seen in a small cohort of postmortem brain samples from patients with schizophrenia and controls. Moreover, in an oligodendrocytic cell line, exposure to MK-801 (dizocilpine, an established model for schizophrenia), antipsychotics, and co-treatments all induced changes in the succinylome and malonylome of these cells, with some changes from MK-801 even being reverted after antipsychotic co-treatment. Despite the limitation that these tissues are not viable targets for widespread biomarkers in personal medicine, this study serves to show that lysine succinylation and malonylation are nonetheless valid and important targets for future biomarker studies, especially in peripheral tissues and fluids. Finally, due to the biological pathways that were enriched by the proteins most affected by these dysregulations, additional studies may also reveal the possible roles of these modifications in the symptoms and treatment of schizophrenia, or the metabolic side effects of antipsychotic medications.

## Figures and Tables

**Figure 1 jpm-12-01408-f001:**
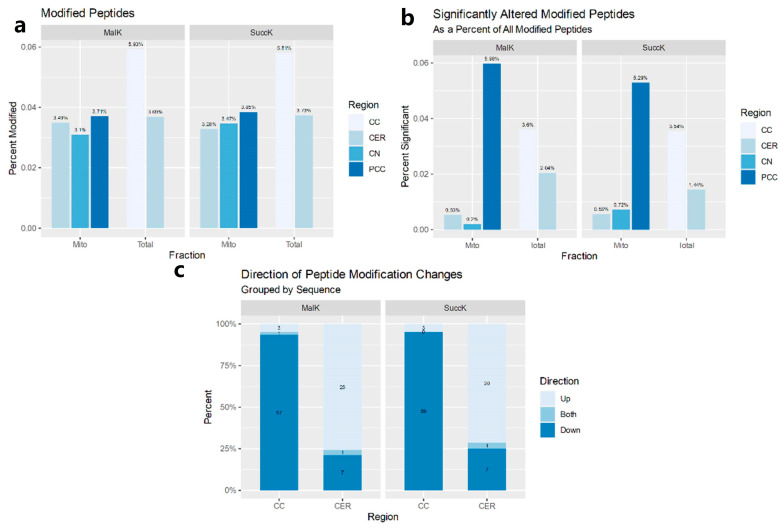
Data involving the relative frequency of the quantified and modified peptides in the posterior cingulate cortex (PCC), corpus callosum (CC), and cerebellum (CER) for succinylation and malonylation: (**a**) The percentage of identified peptides that were modified in mitochondrion-enriched fractions, left, and those that were dysregulated (ANOVA ≤ 0.05), right; (**b**) the percentage of modifications that were increased or decreased (ANOVA ≤ 0.05) in the total cell lysates of the corpus callosum and cerebellum—accession numbers that had different sites found with increased and decreased levels of a modified peptide were labeled as Both; (**c**) the total numbers of succinylation and malonylation sites in the full cell lysates of the corpus callosum and cerebellum, also indicating the percent of these sites that differed in SCZ (ANOVA ≤ 0.05). Posterior cingulate cortex—PCC; corpus callosum—CC; cerebellum—CER; SuccK—succinyllysine; MalK—malonyllysine.

**Figure 2 jpm-12-01408-f002:**
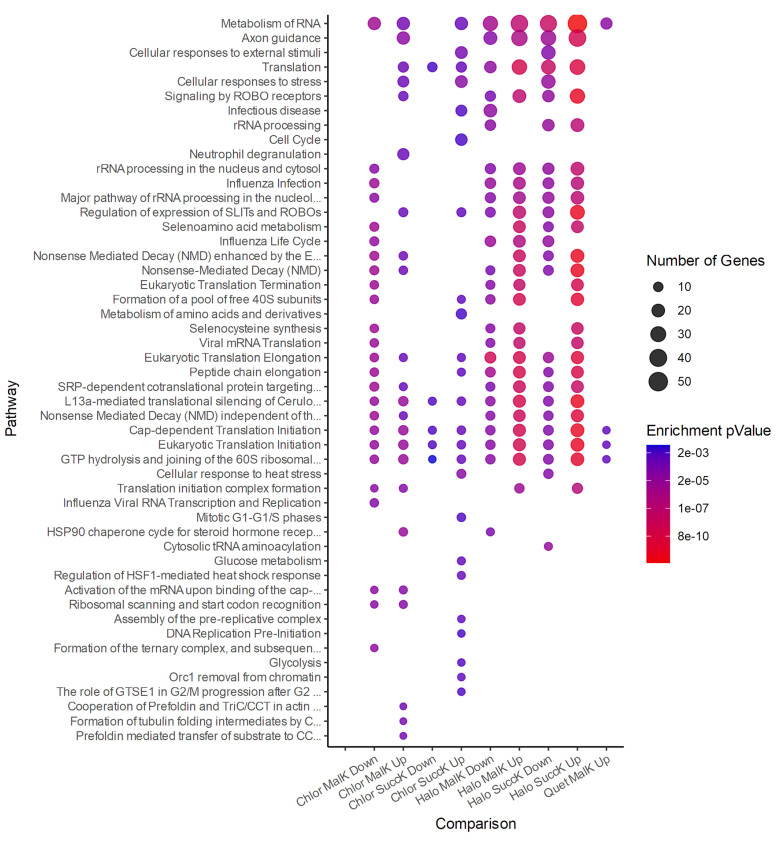
Visualization of biological pathways enriched by proteins found with higher (Up) or lower (Down) levels of succinylation (SuccK) or malonylation (MalK), induced by haloperidol (Halo), chlorpromazine (Chlor), and quetiapine (Quet) exposure in MO3.13 cells, determined by the Reactome.org database. The number of gene names for each enriched pathway determines the size of the dot, while the colors represent the FDR for the pathways. Comparisons not listed did not enrich for any pathways with FDR < 1% and pathways containing fewer than a total of 5 gene names were excluded.

**Table 1 jpm-12-01408-t001:** Sites of succinylation (SuccK) and malonylation (MalK) deregulated by MK-801 (dizocilpine; 4 h) and attenuated by haloperidol, chlorpromazine, and quetiapine (co-exposure, 4 additional hours). Attenuation was classified by a return to under ±10%, ±25%, and ±50% of control levels after dysregulation past that threshold. Proteins marked with asterisks exhibited sites that were dysregulated past one (*) or two (**) additional thresholds before attenuation. All genes are represented by their UniProt gene name.

	Haloperidol			Chlorpromazine			Quetiapine		
	<10%	10–25%	25–50%	<10%	10–25%	25–50%	<10%	10–25%	25–50%
**SuccK**	SNRNP70 **	HSP90AA1 *	RPF2	MATR3 **	NPM1 *	ACLY	CNP **	HSPD1 *	ACTBL2
	DHX9 *	RPS3A *		ACLY *	RDX *	CPT2	RPS15 **	PARP1 *	BCLAF1
	FRMPD2	SEPT11 *		MAPRE1	BTF3	ITGB1	ILF3 *	RAB5C *	DDX49
	TOP2A					LRPPRC ^#^	PTBP3		DDX5
						NACA			DYNC1H1
						PCNA			FIP1L1
						PKM			HSPA7
						SLIRP			NEFM
						TOP1			NUP62
									RPS10
									TUBB
**MalK**	CCT2 *	ALDOB *	ABCE1	ACLY **	API5 *	DSTN	CLTCL1 **	POLDIP3 *	ARF1
	KATNAL2 *	PRKCSH *	DLG3	ESD **	HNRNPLL *	HSPA4	GNL3 **		BUD31
	RPLP0 *	PPP2R1A		HIST1H1E **	RPL27A *	HSPA7			COPA
	BANF1	RAB7A		DDX21	RUVBL2 *	LDHC			EMG1
	SEC61A1			HSP90B1	ACO2	MYH13			HNRNPU
	TBC1D4				HNRNPM	PHGDH			MYL6
					HNRNPU	POTEKP			SLC3A2
					HSP90B1				SPIN3
					RPS14				

^#^ Two different modified peptides were attenuated in the protein LRPPRC.

## Data Availability

The mass spectrometry proteomics data have been deposited to the ProteomeXchange Consortium via the PRIDE partner repository with the dataset identifier PXD014188.

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
