# Peer review of "Protein Succinylation and Malonylation as Potential Biomarkers in Schizophrenia"

_jpm, 2022, doi:10.3390/jpm12091408_

Round 1

Reviewer 1 Report

The authors have clearly presented the need to look into many potential metabolism markers by means of looking for changing in various proteins in PCC and CR and CER regions of the brain through this study. The use of both in-vivo (post mortem) and the in-vitro M03.13 cells covers the basis of the ability of testing the effect of TAPs and AAPs and studying the effect antagonist mediated SCZ and attenuation using anti-pyschotics drugs. 

1) The color scheme used in figure 1 is not clear. Would recommend changing the color to highlight the change. The image seems to grainy. Please update with a more clear image. Same picture clarity issue with Figure 2

2) Some of the full forms of the acronyms are required, Eg- TUFM

3) Are there any common markers that show changes in both succinylation and malonylation post attenuation? If yes, would recommed displaying the commonality or differences by means of a venn diagram 

Author Response

1) Thank you for bringing this to our attention. We updated the figures to make them more easily readable. Regarding image quality, the figures were compressed when added to the document and this has since been resolved.

2) Protein acronyms in the body of the text were all explained. The large list of protein names in the table were labeled as Uniprot gene names.

3) Unfortunately, the number of attenuated proteins in both succinylation and malonylation were very low. In fact, only 3 proteins were found to be attenuated in more than one condition. From a biological standpoint this is low; however, we hope this serves as a possible proof of concept for more elaborate future biomarker or pathway studies.

We also noticed that a lower score was given to the methods section, which was improved in response on the other reviewer's comments.

Reviewer 2 Report

The authors investigated protein succinylation and malonylation as potential biomarkers in schizophrenia, which has important research value in the treatment of schizophrenia. The experimental design of this study is reasonable and meaningful. However, there are some minor details in the article that need to be revised. The article can be published after a minor revision.

The specific content is as follows:

1.      It is suggested that the subheading "1.1, 1.2 and 1.3" be added to the "1. Introduction" section. For example, 1.1. Schizophrenia and Antipsychotics. In addition, the author should briefly introduce some of these three subsections in the first paragraph of the "Introduction Section", and then proceed to introduce them separately.

2.      2. Materials and Methods, The same problem occurred and suggested to add the title serial number. For example, 2.1. Postmortem Tissue Preparation

3.      Liquid Chromatography-Mass Spectrometry. Please provide the specific model, company and other information of the instrument, as well as specific LC and MS conditions.

4.      3. Results, The clarity of figure 1 and 2 is not enough, please adjust it.

5.      The table 1 frame looks strange, please adjust it.

6.      A table below "5. Conclusion" is not explained, which is puzzling.

Author Response

Thank you for your comments and suggestions. We will respond to each point below. Slight modifications were made to the results section to better accommodate Figure 2. The overall results do not change because of this.

1) We agree with the suggestion to separate the introduction into individual sections. A brief introduction to these individual sections, however, seemed redundant without removing some details from the other section.

2) This section is now numbered and separated.

3) The instrument information and overarching instrument configurations were included. Care was taken to make it clear which citations contain more specific data for each previously carried out analysis.

4) The figures were recreated for better readability and overall quality. Undesired image compression in docx files was corrected.

5) We appreciate the feedback. The model for the journal was used for the table; however, our data format is not entirely compatible. We added dotted lines and more weighted lines to better divide the sections in the table for clarity.

6) This table was an older version of Table 1 that somehow did not get removed from the final version before sending. We apologize for the confusion.